# ENTROPY-SGD: BIASING GRADIENT DESCENT INTO WIDE VALLEYS

**Pratik Chaudhari**[1], **Anna Choromanska**[2], **Stefano Soatto**[1], **Yann LeCun**[3,4], **Carlo Baldassi**[5], **Christian Borgs**[6], **Jennifer Chayes**[6], **Levent Sagun**[3], **Riccardo Zecchina**[5]

[1] Computer Science Department, University of California, Los Angeles

[2] Department of Electrical and Computer Engineering, New York University

[3] Courant Institute of Mathematical Sciences, New York University

[4] Facebook AI Research, New York

[5] Dipartimento di Scienza Applicata e Tecnologia, Politecnico di Torino

[6] Microsoft Research New England, Cambridge

Email: pratikac@ucla.edu, ac5455@nyu.edu, soatto@ucla.edu, yann@cs.nyu.edu, carlo.baldassi@polito.it, borgs@microsoft.com, jchayes@microsoft.com, sagun@cims.nyu.edu, riccardo.zecchina@polito.it

## ABSTRACT

This paper proposes a new optimization algorithm called Entropy-SGD for training deep neural networks that is motivated by the local geometry of the energy landscape. Local extrema with low generalization error have a large proportion of almost-zero eigenvalues in the Hessian with very few positive or negative eigenvalues. We leverage upon this observation to construct a local-entropy-based objective function that favors well-generalizable solutions lying in large flat regions of the energy landscape, while avoiding poorly-generalizable solutions located in the sharp valleys. Conceptually, our algorithm resembles two nested loops of SGD where we use Langevin dynamics in the inner loop to compute the gradient of the local entropy before each update of the weights. We show that the new objective has a smoother energy landscape and show improved generalization over SGD using uniform stability, under certain assumptions. Our experiments on convolutional and recurrent neural networks demonstrate that Entropy-SGD compares favorably to state-of-the-art techniques in terms of generalization error and training time.

## 1 INTRODUCTION

This paper presents a new optimization tool for deep learning designed to exploit the local geometric properties of the objective function. Consider the histogram we obtained in Fig. 1 showing the spectrum of the Hessian at an extremum discovered by Adam (Kingma & Ba, 2014) for a convolutional neural network on MNIST (LeCun et al., 1998) ($\approx 47,000$ weights, cf. Sec. 5.1). It is evident that:

(i) a large number of directions ($\approx 94\%$) have near-zero eigenvalues (magnitude less than $10^{-4}$),

(ii) positive eigenvalues (right inset) have a long tail with the largest one being almost 40,

(iii) negative eigenvalues (left inset), which are directions of descent that the optimizer missed, have a much faster decay (the largest negative eigenvalue is only $-0.46$).

Interestingly, this trend is not unique to this particular network. Rather, its qualitative properties are shared across a variety of network architectures, network sizes, datasets or optimization algorithms (refer to Sec. 5 for more experiments). Local minima that generalize well and are discovered by gradient descent lie in "wide valleys" of the energy landscape, rather than in sharp, isolated minima. For an intuitive understanding of this phenomenon, imagine a Bayesian prior concentrated at the minimizer of the expected loss, the marginal likelihood of wide valleys under this prior is much

higher than narrow, sharp valleys even if the latter are close to the global minimum in training loss. Almost-flat regions of the energy landscape are robust to data perturbations, noise in the activations, as well as perturbations of the parameters, all of which are widely-used techniques to achieve good generalization. This suggests that wide valleys should result in better generalization and, indeed, standard optimization algorithms in deep learning seem to discover exactly that — without being explicitly tailored to do so. For another recent analysis of the Hessian, see the parallel work of Sagun et al. (2016).

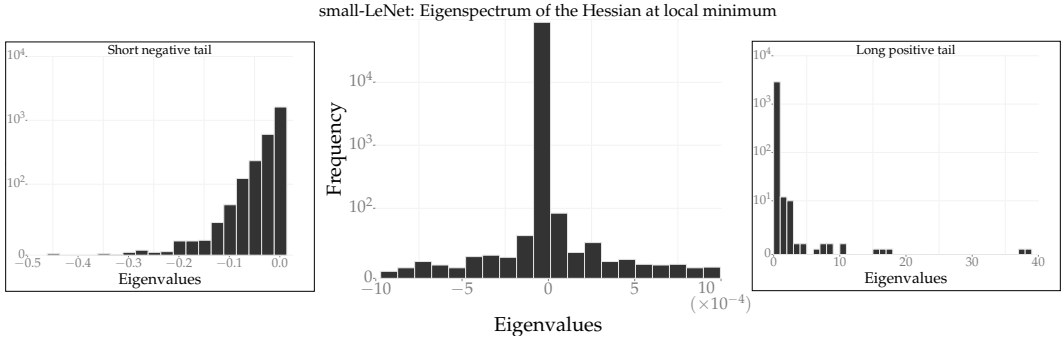

Figure 1: Eigenspectrum of the Hessian at a local minimum of a CNN on MNIST (two independent runs). **Remark:** The central plot shows the eigenvalues in a small neighborhood of zero whereas the left and right insets show the entire tails of the eigenspectrum.

Based on this understanding of how the local geometry looks at the end of optimization, can we modify SGD to actively seek such regions? Motivated by the work of Baldassi et al. (2015) on shallow networks, instead of minimizing the original loss $f(x)$, we propose to maximize

$$F(x, \gamma) = \log \int_{x' \in \mathbb{R}^n} \exp\left(-f(x') - \frac{\gamma}{2} \|x - x'\|_2^2\right) \, dx'.$$

The above is a log-partition function that measures both the depth of a valley at a location $x \in \mathbb{R}^n$, and its flatness through the entropy of $f(x')$; we call it "local entropy" in analogy to the free entropy used in statistical physics. The Entropy-SGD algorithm presented in this paper employs stochastic gradient Langevin dynamics (SGLD) to approximate the gradient of local entropy. Our algorithm resembles two nested loops of SGD: the inner loop consists of SGLD iterations while the outer loop updates the parameters. We show that the above modified loss function results in a smoother energy landscape defined by the hyper-parameter $\gamma$ which we can think of as a "scope" that seeks out valleys of specific widths. Actively biasing the optimization towards wide valleys in the energy landscape results in better generalization error. We present experimental results on fully-connected and convolutional neural networks (CNNs) on the MNIST and CIFAR-10 (Krizhevsky, 2009) datasets and recurrent neural networks (RNNs) on the Penn Tree Bank dataset (PTB) (Marcus et al., 1993) and character-level text prediction. Our experiments show that Entropy-SGD scales to deep networks used in practice, obtains comparable generalization error as competitive baselines and also trains much more quickly than SGD (we get a 2x speed-up over SGD on RNNs).

## 2 RELATED WORK

Our above observation about the spectrum of Hessian (further discussed in Sec. 5) is similar to results on a perceptron model in Dauphin et al. (2014) where the authors connect the loss function of a deep network to a high-dimensional Gaussian random field. They also relate to earlier studies such as Baldi & Hornik (1989); Fyodorov & Williams (2007); Bray & Dean (2007) which show that critical points with high training error are exponentially likely to be saddle points with many negative directions and all local minima are likely to have error that is very close to that of the global minimum. The authors also argue that convergence of gradient descent is affected by the proliferation of saddle points surrounded by high error plateaus — as opposed to multiple local minima. One can also see this via an application of Kramer's law: the time spent by diffusion is

inversely proportional to the smallest negative eigenvalue of the Hessian at a saddle point (Bovier & den Hollander, 2006).

The existence of multiple — almost equivalent — local minima in deep networks has been predicted using a wide variety of theoretical analyses and empirical observations, e.g., papers such as Choromanska et al. (2015a;b); Chaudhari & Soatto (2015) that build upon results from statistical physics as also others such as Haeffele & Vidal (2015) and Janzamin et al. (2015) that obtain similar results for matrix and tensor factorization problems. Although assumptions in these works are somewhat unrealistic in the context of deep networks used in practice, similar results are also true for linear networks which afford a more thorough analytical treatment (Saxe et al., 2014). For instance, Soudry & Carmon (2016) show that with mild over-parameterization and dropout-like noise, training error for a neural network with one hidden layer and piece-wise linear activation is zero at every local minimum. All these results suggest that the energy landscape of deep neural networks should be easy to optimize and they more or less hold in practice — it is easy to optimize a prototypical deep network to near-zero loss *on the training set* (Hardt et al., 2015; Goodfellow & Vinyals, 2015).

Obtaining good *generalization* error, however, is challenging: complex architectures are sensitive to initial conditions and learning rates (Sutskever et al., 2013) and even linear networks (Kawaguchi, 2016) may have degenerate and hard to escape saddle points (Ge et al., 2015; Anandkumar & Ge, 2016). Techniques such as adaptive (Duchi et al., 2011) and annealed learning rates, momentum (Tieleman & Hinton, 2012), as well as architectural modifications like dropout (Srivastava et al., 2014), batch-normalization (Ioffe & Szegedy, 2015; Cooijmans et al., 2016), weight scaling (Salimans & Kingma, 2016) etc. are different ways of tackling this issue by making the underlying landscape more amenable to first-order algorithms. However, the training process often requires a combination of such techniques and it is unclear beforehand to what extent each one of them helps.

Closer to the subject of this paper are results by Baldassi et al. (2015; 2016a;b) who show that the energy landscape of shallow networks with discrete weights is characterized by an exponential number of isolated minima and few very dense regions with lots of local minima close to each other. These dense local minima can be shown to generalize well for random input data; more importantly, they are also accessible by efficient algorithms using a novel measure called "robust ensemble" that amplifies the weight of such dense regions. The authors use belief propagation to estimate local entropy for simpler models such as committee machines considered there. A related work in this context is EASGD (Zhang et al., 2015) which trains multiple deep networks in parallel and modulates the distance of each worker from the ensemble average. Such an ensemble training procedure enables improved generalization by ensuring that different workers land in the same wide valley and indeed, it turns out to be closely related to the replica theoretic analysis of Baldassi et al. (2016a).

Our work generalizes the local entropy approach above to modern deep networks with continuous weights. It exploits the same phenomenon of wide valleys in the energy landscape but does so without incurring the hardware and communication complexity of replicated training or being limited to models where one can estimate local entropy using belief propagation. The enabling technique in our case is using Langevin dynamics for estimating the gradient of local entropy, which can be done efficiently even for large deep networks using mini-batch updates.

Motivated by the same final goal, viz. flat local minima, the authors in Hochreiter & Schmidhuber (1997b) introduce hard constraints on the training loss and the width of local minima and show using the Gibbs formalism (Haussler & Opper, 1997) that this leads to improved generalization. As the authors discuss, the effect of hyper-parameters for the constraints is intricately tied together and they are difficult to choose even for small problems. Our local entropy based objective instead naturally balances the energetic term (training loss) and the entropic term (width of the valley). The role of $\gamma$ is clear as a focusing parameter (cf. Sec. 4.3) and effectively exploiting this provides significant computational advantages. Among other conceptual similarities with our work, let us note that local entropy in a flat valley is a direct measure of the width of the valley which is similar to their usage of Hessian, while the Gibbs variant to averaging in weight space (Eqn. 33 of Hochreiter & Schmidhuber (1997b)) is similar to our Eqn. (7). Indeed, Gibbs formalism used in their analysis is a very promising direction to understanding generalization in deep networks (Zhang et al., 2016).

Homotopy continuation methods convolve the loss function to solve sequentially refined optimization problems (Allgower & Georg, 2012; Mobahi & Fisher III, 2015), similarly, methods that perturb the weights or activations to average the gradient (Gulcehre et al., 2016) do so with an aim to smooth the rugged energy landscape. Such smoothing is however very different from local entropy. For instance, the latter places more weight on wide local minima even if they are much shallower than the global minimum (cf. Fig. 2); this effect cannot be obtained by smoothing. In fact, smoothing can introduce an artificial minimum between two nearby sharp valleys which is detrimental to generalization. In order to be effective, continuation techniques also require that minimizers of successively smaller convolutions of the loss function lie close to each other (Hazan et al., 2016); it is not clear whether this is true for deep networks. Local entropy, on the other hand, exploits wide minima which have been shown to exist in a variety of learning problems (Monasson & Zecchina, 1995; Cocco et al., 1996). Please refer to Appendix C for a more elaborate discussion as well as possible connections to stochastic variational inference (Blei et al., 2016).

## 3 LOCAL ENTROPY

We first provide a simple intuition for the concept of local entropy of an energy landscape. The discussion in this section builds upon the results of Baldassi et al. (2016a) and extends it for the case of continuous variables. Consider a cartoon energy landscape in Fig. 2 where the x-axis denotes the configuration space of the parameters. We have constructed two local minima: a shallower although wider one at $x_{\text{robust}}$ and a very sharp global minimum at $x_{\text{non-robust}}$. Under a Bayesian prior on the parameters, say a Gaussian of a fixed variance at locations $x_{\text{robust}}$ and $x_{\text{non-robust}}$ respectively, the wider local minimum has a higher marginalized likelihood than the sharp valley on the right.

The above discussion suggests that parameters that lie in wider local minima like $x_{\text{robust}}$, which may possibly have a higher loss than the global minimum, should generalize better than the ones that are simply at the global minimum. In a neighborhood of $x_{\text{robust}}$, "local entropy" as introduced in Sec. 1 is large because it includes the contributions from a large region of good parameters; conversely, near $x_{\text{non-robust}}$, there are fewer such contributions and the resulting local entropy is low. The local entropy thus provides a way of picking large, approximately flat, regions of the landscape over sharp, narrow valleys in spite of the latter possibly having a lower loss. Quite conveniently, the local entropy is also computed from the partition function with a local re-weighting term.

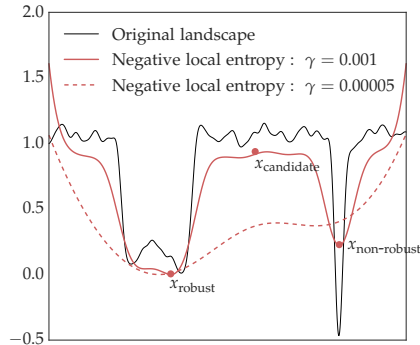

Figure 2: Local entropy concentrates on wide valleys in the energy landscape.

Formally, for a parameter vector $x \in \mathbb{R}^n$, consider a Gibbs distribution corresponding to a given energy landscape $f(x)$:

$$P(x;\ \beta) = Z_\beta^{-1}\ \exp\ (-\beta\ f(x));\qquad(1)$$

where $\beta$ is known as the inverse temperature and $Z_\beta$ is a normalizing constant, also known as the partition function. As $\beta \to \infty$, the probability distribution above concentrates on the global minimum of $f(x)$ (assuming it is unique) given as:

$$x^* = \underset{x}{\text{argmin}}\ f(x),\qquad(2)$$

which establishes the link between the Gibbs distribution and a generic optimization problem (2). We would instead like the probability distribution — and therefore the underlying optimization problem — to focus on flat regions such as $x_{\text{robust}}$ in Fig. 2. With this in mind, let us construct a modified Gibbs distribution:

$$P(x';\ x,\beta,\gamma) = Z_{x,\beta,\ \gamma}^{-1}\ \exp\ \left(-\beta\ f(x') - \beta\ \frac{\gamma}{2}\ \|x-x'\|_2^2\right).\qquad(3)$$

The distribution in (3) is a function of a dummy variable $x'$ and is parameterized by the original location $x$. The parameter $\gamma$ biases the modified distribution (3) towards $x$; a large $\gamma$ results in a

$P(x'; x, \beta, \gamma)$ with all its mass near $x$ irrespective of the energy term $f(x')$. For small values of $\gamma$, the term $f(x')$ in the exponent dominates and the modified distribution is similar to the original Gibbs distribution in (1). We will set the inverse temperature $\beta$ to 1 because $\gamma$ affords us similar control on the Gibbs distribution.

**Definition 1** (**Local entropy**). The local free entropy of the Gibbs distribution in (1), colloquially called "local entropy" in the sequel and denoted by $F(x, \gamma)$, is defined as the log-partition function of modified Gibbs distribution (3), i.e.,

$$
\begin{aligned}
F(x, \gamma) &= \log Z_{x,1,\gamma} \\
&= \log \int_{x'} \exp\left(-f(x') - \frac{\gamma}{2}\|x - x'\|_2^2\right) dx'.
\end{aligned}
\tag{4}
$$

The parameter $\gamma$ is used to focus the modified Gibbs distribution upon a local neighborhood of $x$ and we call it a "scope" henceforth.

**Effect on the energy landscape:** Fig. 2 shows the negative local entropy $-F(x, \gamma)$ for two different values of $\gamma$. We expect $x_{\text{robust}}$ to be more robust than $x_{\text{non-robust}}$ to perturbations of data or parameters and thus generalize well and indeed, the negative local entropy in Fig. 2 has a **global** minimum near $x_{\text{robust}}$. For low values of $\gamma$, the energy landscape is significantly smoother than the original landscape and still maintains our desired characteristic, viz. global minimum at a wide valley. As $\gamma$ increases, the local entropy energy landscape gets closer to the original energy landscape and they become equivalent at $\gamma \to \infty$. On the other hand, for very small values of $\gamma$, the local entropy energy landscape is almost uniform.

**Connection to classical entropy:** The quantity we have defined as local entropy in Def. 1 is different from classical entropy which counts the number of likely configurations under a given distribution. For a continuous parameter space, this is given by

$$
S(x, \beta, \gamma) = -\int_{x'} \log P(x'; x, \beta, \gamma) \, dP(x'; x, \beta, \gamma)
$$

for the Gibbs distribution in (3). Minimizing classical entropy however does not differentiate between flat regions that have very high loss versus flat regions that lie deeper in the energy landscape. For instance in Fig. 2, classical entropy is smallest in the neighborhood of $x_{\text{candidate}}$ which is a large region with very high loss on the training dataset and is unlikely to generalize well.

## 4 ENTROPY-GUIDED SGD

Simply speaking, our Entropy-SGD algorithm minimizes the negative local entropy from Sec. 3. This section discusses how the gradient of local entropy can be computed via Langevin dynamics. The reader will see that the resulting algorithm has a strong flavor of "SGD-inside-SGD": the outer SGD updates the parameters, while an inner SGD estimates the gradient of local entropy.

Consider a typical classification setting, let $x \in \mathbb{R}^n$ be the weights of a deep neural network and $\xi_k \in \Xi$ be samples from a dataset $\Xi$ of size $N$. Let $f(x; \xi_k)$ be the loss function, e.g., cross-entropy of the classifier on a sample $\xi_k$. The original optimization problem is:

$$
x^* = \underset{x}{\text{argmin}} \ \frac{1}{N} \sum_{k=1}^{N} f(x; \xi_k);
\tag{5}
$$

where the objective $f(x, \xi_k)$ is typically a non-convex function in both the weights $x$ and the samples $\xi_k$. The Entropy-SGD algorithm instead solves the problem

$$
x_{\text{Entropy-SGD}}^* = \underset{x}{\text{argmin}} \ -F(x, \gamma; \Xi);
\tag{6}
$$

where we have made the dependence of local entropy $F(x, \gamma)$ on the dataset $\Xi$ explicit.

### 4.1 Gradient of local entropy

The gradient of local entropy over a randomly sampled mini-batch of $m$ samples denoted by $\xi_{\ell_i} \in \Xi^\ell$ for $i \leq m$ is easy to derive and is given by

$$-\nabla_x F\left(x, \gamma; \Xi^\ell\right) = \gamma\left(x - \left\langle x'; \Xi^\ell \right\rangle\right); \qquad (7)$$

where the notation $\langle \cdot \rangle$ denotes an expectation of its arguments (we have again made the dependence on the data explicit) over a Gibbs distribution of the original optimization problem modified to focus on the neighborhood of $x$; this is given by

$$P(x'; x, \gamma) \propto \exp\left[-\left(\frac{1}{m}\sum_{i=1}^{m} f\left(x'; \xi_{\ell_i}\right)\right) - \frac{\gamma}{2} \|x - x'\|_2^2\right]. \qquad (8)$$

Computationally, the gradient in (7) involves estimating $\langle x'; \Xi^\ell \rangle$ with the current weights fixed to $x$. This is an expectation over a Gibbs distribution and is hard to compute. We can however approximate it using Markov chain Monte-Carlo (MCMC) techniques. In this paper, we use stochastic gradient Langevin dynamics (SGLD) (Welling & Teh, 2011) that is an MCMC algorithm for drawing samples from a Bayesian posterior and scales to large datasets using mini-batch updates. Please see Appendix A for a brief overview of SGLD. For our application, as lines 3-6 of Alg. 1 show, SGLD resembles a few iterations of SGD with a forcing term $-\gamma(x - x')$ and additive gradient noise.

We can obtain some intuition on how Entropy-SGD works using the expression for the gradient: the term $\langle x'; \cdot \rangle$ is the average over a locally focused Gibbs distribution and for two local minima in the neighborhood of $x$ roughly equivalent in loss, this term points towards the wider one because $\langle x'; \cdot \rangle$ is closer to it. This results in a net gradient that takes SGD towards wider valleys. Moreover, if we unroll the SGLD steps used to compute $(x - \langle x'; \cdot \rangle)$ (cf. line 5 in Alg. 1), it resembles one large step in the direction of the (noisy) average gradient around the current weights $x$ and Entropy-SGD thus looks similar to averaged SGD in the literature (Polyak & Juditsky, 1992; Bottou, 2012). These two phenomena intuitively explain the improved generalization performance of Entropy-SGD.

### 4.2 Algorithm and Implementation details

Alg. 1 provides the pseudo-code for one iteration of the Entropy-SGD algorithm. At each iteration, lines 3-6 perform $L$ iterations of Langevin dynamics to estimate $\mu = \langle x'; \Xi^\ell \rangle$. The weights $x$ are updated with the modified gradient on line 7.

---

**Algorithm 1:** Entropy-SGD algorithm

|  |  |  |
|---|---|---|
| **Input** | : | current weights $x$, Langevin iterations $L$ |
| **Hyper-parameters:** |  | scope $\gamma$, learning rate $\eta$, SGLD step size $\eta'$ |

// SGLD iterations;

1  $x', \mu \leftarrow x$;

2  **for** $\ell \leq L$ **do**

3  $\quad$ $\Xi^\ell \leftarrow$ sample mini-batch;

4  $\quad$ $dx' \leftarrow \frac{1}{m}\sum_{i=1}^{m} \nabla_{x'} f\left(x'; \xi_{\ell_i}\right) - \gamma\left(x - x'\right)$;

5  $\quad$ $x' \leftarrow x' - \eta'\, dx' + \sqrt{\eta'}\, \varepsilon\, N(0, I)$;

6  $\quad$ $\mu \leftarrow (1 - \alpha)\mu + \alpha\, x'$;

// Update weights;

7  $x \leftarrow x - \eta\,\gamma\,(x - \mu)$

---

Let us now discuss a few implementation details. Although we have written Alg. 1 in the classical SGD setup, we can easily modify it to include techniques such as momentum and gradient preconditioning (Duchi et al., 2011) by changing lines 5 and 7. In our experiments, we have used SGD with Nesterov's momentum (Sutskever et al., 2013) and Adam for outer and inner loops with similar qualitative results. We use exponential averaging to estimate $\mu$ in the SGLD loop (line 6)

with $\alpha = 0.75$ so as to put more weight on the later samples, this is akin to a burn-in in standard SGLD.

We set the number of SGLD iterations to $L = [5, 20]$ depending upon the complexity of the dataset. The learning rate $\eta'$ is fixed for all our experiments to values between $\eta' \in [0.1, 1]$. We found that annealing $\eta'$ (for instance, setting it to be the same as the outer learning rate $\eta$) is detrimental; indeed a small learning rate leads to poor estimates of local entropy towards the end of training where they are most needed. The parameter $\varepsilon$ in SGLD on line 5 is the thermal noise and we fix this to $\varepsilon \in [10^{-4}, 10^{-3}]$. Having thus fixed $L, \varepsilon$ and $\eta'$, an effective heuristic to tune the remaining parameter $\gamma$ is to match the magnitude of the gradient of the local entropy term, viz. $\gamma (x - \mu)$, to the gradient for vanilla SGD, viz. $m^{-1} \sum_{i=1}^{m} \nabla_x f(x; \xi_{\ell_i})$.

### 4.3 SCOPING OF $\gamma$

The scope $\gamma$ is fixed in Alg. 1. For large values of $\gamma$, the SGLD updates happen in a small neighborhood of the current parameters $x$ while low values of $\gamma$ allow the inner SGLD to explore further away from $x$. In the context of the discussion in Sec. 3, a "reverse-annealing" of the scope $\gamma$, i.e. increasing $\gamma$ as training progresses has the effect of exploring the energy landscape on progressively finer scales. We call this process "scoping" which is similar to that of Baldassi et al. (2016a) and use a simple exponential schedule given by

$$\gamma(t) = \gamma_0 \ (1 + \gamma_1)^t;$$

for the $t^{\text{th}}$ parameter update. We have experimented with linear, quadratic and bounded exponential $(\gamma_0 (1 - e^{-\tau t}))$ scoping schedules and obtained qualitatively similar results.

Scoping of $\gamma$ unfortunately interferes with the learning rate annealing that is popular in deep learning, this is a direct consequence of the update step on line 7 of Alg. 1. In practice, we therefore scale down the local entropy gradient by $\gamma$ before the weight update and modify the line to read

$$x \leftarrow x - \eta (x - \mu).$$

Our general strategy during hyper-parameter tuning is to set the initial scope $\gamma_0$ to be very small, pick a large value of $\eta$ and set $\gamma_1$ to be such that the magnitude of the local entropy gradient is comparable to that of SGD. We can use much larger learning rates than SGD in our experiments because the local entropy gradient is less noisy than the original back-propagated gradient. This also enables very fast progress in the beginning with a smooth landscape of a small $\gamma$.

### 4.4 THEORETICAL PROPERTIES

We can show that Entropy-SGD results in a smoother loss function and obtains better generalization error than the original objective (5). With some overload of notation, we assume that the original loss $f(x)$ is $\beta$-smooth, i.e., for all $x, y \in \mathbb{R}^n$, we have $\|\nabla f(x) - \nabla f(y)\| \leq \beta \|x - y\|$. We additionally assume for the purpose of analysis that no eigenvalue of the Hessian $\nabla^2 f(x)$ lies in the set $[-2\gamma - c, c]$ for some small $c > 0$.

**Lemma 2.** *The objective $F(x, \gamma; \Xi)$ in (6) is $\frac{\alpha}{1+\gamma^{-1} c}$-Lipschitz and $\frac{\beta}{1+\gamma^{-1} c}$-smooth.*

Please see Appendix B for the proof. The local entropy objective is thus smoother than the original objective. Let us now obtain a bound on the improvement in generalization error. We denote an optimization algorithm, viz., SGD or Entropy-SGD by $A(\Xi)$, it is a function of the dataset $\Xi$ and outputs the parameters $x^*$ upon termination. Stability of the algorithm (Bousquet & Elisseeff, 2002) is then a notion of how much its output differs in loss upon being presented with two datasets, $\Xi$ and $\Xi'$, that differ in at most one sample:

$$\sup_{\xi \ \in \ \Xi \cup \Xi'} \left[ f(A(\Xi), \xi) - f(A(\Xi'), \xi) \right] \leq \varepsilon.$$

Hardt et al. (2015) connect uniform stability to generalization error and show that an $\varepsilon$-stable algorithm $A(\Xi)$ has generalization error bounded by $\varepsilon$, i.e., if $A(\Xi)$ terminates with parameters $x^*$,

$$|\mathbb{E}_\Xi (R_\Xi(x^*) - R(x^*))| \leq \varepsilon;$$

where the left hand side is the generalization error: it is the difference between the empirical loss $R_{\Xi}(x) := N^{-1} \sum_{k=1}^{N} f(x, \xi_k)$ and the population loss $R(x) := \mathbb{E}_{\xi} f(x, \xi)$. We now employ the following theorem that bounds the stability of an optimization algorithm through the smoothness of its loss function and the number of iterations on the training set.

**Theorem 3** (Hardt et al. (2015)). *For an $\alpha$-Lipschitz and $\beta$-smooth loss function, if SGD converges in $T$ iterations on $N$ samples with decreasing learning rate $\eta_t \leq 1/t$ the stability is bounded by*

$$\varepsilon \lesssim \frac{1}{N} \, \alpha^{1/(1+\beta)} \, T^{1-1/(1+\beta)}.$$

Using Lemma 2 and Theorem 3 we have

$$\varepsilon_{\text{Entropy-SGD}} \lesssim \left( \alpha \, T^{-1} \right)^{\left( 1 - \frac{1}{1+\gamma^{-1}c} \right) \beta} \, \varepsilon_{\text{SGD}}, \tag{9}$$

which shows that Entropy-SGD generalizes better than SGD for all $T > \alpha$ if they both converge after $T$ passes over the samples.

Let us note that while the number of passes over the dataset for Entropy-SGD and SGD are similar for our experiments on CNNs, Entropy-SGD makes only half as many passes as SGD for our experiments on RNNs. As an aside, it is easy to see from the proof of Lemma 2 that for a convex loss function $f(x)$, the local entropy objective does not change the minimizer of the original problem.

**Remark 4.** The above analysis hinges upon an assumption that the Hessian $\nabla^2 f(x)$ does not have eigenvalues in the set $[-2\gamma - c, c]$ for a constant $c > 0$. This is admittedly unrealistic, for instance, the eigenspectrum of the Hessian at a local minimum in Fig. 1 has a large fraction of its eigenvalues almost zero. Let us however remark that the result in Thm. 3 by Hardt et al. (2015) assumes global conditions on the smoothness of the loss function; one imagines that Eqn. 9 remains qualitatively the same (with respect to $T$ in particular) even if this assumption is violated to an extent before convergence happens. Obtaining a rigorous generalization bound without this assumption would require a dynamical analysis of SGD and seems out of reach currently.

## 5 EXPERIMENTS

In Sec. 5.1, we discuss experiments that suggest that the characteristics of the energy landscape around local minimal accessible by SGD are universal to deep architectures. We then present experimental results on two standard image classification datasets, viz. MNIST and CIFAR-10 and two datasets for text prediction, viz. PTB and the text of War and Peace. Table 1 summarizes the results of these experiments on deep networks.

### 5.1 UNIVERSALITY OF THE HESSIAN AT LOCAL MINIMA

We use automatic differentiation[1] to compute the Hessian at a local minimum obtained at the end of training for the following networks:

(i) **small-LeNet on MNIST**: This network has $47,658$ parameters and is similar to LeNet but with 10 and 20 channels respectively in the first two convolutional layers and 128 hidden units in the fully-connected layer. We train this with Adam to obtain a test error of 2.4%.

(ii) **small-mnistfc on MNIST**: A fully-connected network ($50,890$ parameters) with one layer of 32 hidden units, ReLU non-linearities and cross-entropy loss; it converges to a test error of 2.5% with momentum-based SGD.

(iii) **char-lstm for text generation**: This is a recurrent network with 48 hidden units and Long Short-Term Memory (LSTM) architecture (Hochreiter & Schmidhuber, 1997a). It has $32,640$ parameters and we train it with Adam to re-generate a small piece of text consisting of 256 lines of length 32 each and 96-bit one-hot encoded characters.

---

[1]https://github.com/HIPS/autograd

(iv) **All-CNN-BN on CIFAR-10**: This is similar to the All-CNN-C network (Springenberg et al., 2014) with $\approx 1.6$ million weights (cf. Sec. 5.3) which we train using Adam to obtain an error of 11.2%. Exact Hessian computation is in this case expensive and thus we instead compute the diagonal of the Fisher information matrix (Wasserman, 2013) using the element-wise first and second moments of the gradients that Adam maintains, i.e., $\text{diag}(I) = \mathbb{E}(g^2) - (\mathbb{E} \ g)^2$ where $g$ is the back-propagated gradient. Fisher information measures the sensitivity of the log-likelihood of data given parameters in a neighborhood of a local minimum and thus is exactly equal to the Hessian of the negative log-likelihood. We will consider the diagonal of the empirical Fisher information matrix as a proxy for the eigenvalues of the Hessian, as is common in the literature.

We choose to compute the exact Hessian and to keep the computational and memory requirements manageable, the first three networks considered above are smaller than standard deep networks used in practice. For the last network, we sacrifice the exact computation and instead approximate the Hessian of a large deep network. We note that recovering an approximate Hessian from Hessian-vector products (Pearlmutter, 1994) could be a viable strategy for large networks.

Fig. 1 in the introductory Sec. 1 shows the eigenspectrum of the Hessian for small-LeNet while Fig. 3 shows the eigenspectra for the other three networks. A large proportion of eigenvalues of the Hessian are very close to zero or positive with a very small (relative) magnitude. This suggests that the local geometry of the energy landscape is almost flat at local minima discovered by gradient descent. This agrees with theoretical results such as Baldassi et al. (2016c) where the authors predict that flat regions of the landscape generalize better. Standard regularizers in deep learning such as convolutions, max-pooling and dropout seem to bias SGD towards flatter regions in the energy landscape. Away from the origin, the right tails of the eigenspectra are much longer than the left tails. Indeed, as discussed in numerous places in literature (Bray & Dean, 2007; Dauphin et al., 2014; Choromanska et al., 2015a), SGD finds low-index critical points, i.e., optimizers with few negative eigenvalues of the Hessian. What is interesting and novel is that the directions of descent that SGD misses do not have a large curvature.

## 5.2 MNIST

We consider two prototypical networks: the first, called mnistfc, is a fully-connected network with two hidden layers of 1024 units each and the second is a convolutional neural network with the same size as LeNet but with batch-normalization (Ioffe & Szegedy, 2015); both use a dropout of probability 0.5 after each layer. As a baseline, we train for 100 epochs with Adam and a learning rate of $10^{-3}$ that drops by a factor of 5 after every 30 epochs to obtain an average error of $1.39 \pm 0.03\%$ and $0.51 \pm 0.01\%$ for mnistfc and LeNet respectively, over 5 independent runs.

For both these networks, we train Entropy-SGD for 5 epochs with $L = 20$ and reduce the dropout probability (0.15 for mnistfc and 0.25 for LeNet). The learning rate of the SGLD updates is fixed to $\eta' = 0.1$ while the outer loop's learning rate is set to $\eta = 1$ and drops by a factor of 10 after the second epoch; we use Nesterov's momentum for both loops. The thermal noise in SGLD updates (line 5 of Alg. 1) is set to $10^{-3}$. We use an exponentially increasing value of $\gamma$ for scoping, the initial value of the scope is set to $\gamma = 10^{-4}$ and this increases by a factor of 1.001 after each parameter update. The results in Fig. 4a and Fig. 4b show that Entropy-SGD obtains a comparable generalization error: $1.37 \pm 0.03\%$ and $0.50 \pm 0.01\%$, for mnistfc and LeNet respectively. While Entropy-SGD trains slightly faster in wall-clock time for LeNet; it is marginally slower for mnistfc which is a small network and trains in about two minutes anyway.

**Remark on the computational complexity:** Since Entropy-SGD runs $L$ steps of SGLD before each parameter update, the effective number of passes over the dataset is $L$ times that of SGD or Adam for the same number of parameter updates. We therefore plot the error curves of Entropy-SGD in Figs. 4, 5, and 6 against the "effective number of epochs", i.e. by multiplying the abscissae by a factor of $L$. (we define $L = 1$ for SGD or Adam). Modulo the time required for the actual parameter updates (which are fewer for Entropy-SGD), this is a direct measure of wall-clock time, agnostic to the underlying hardware and implementation.

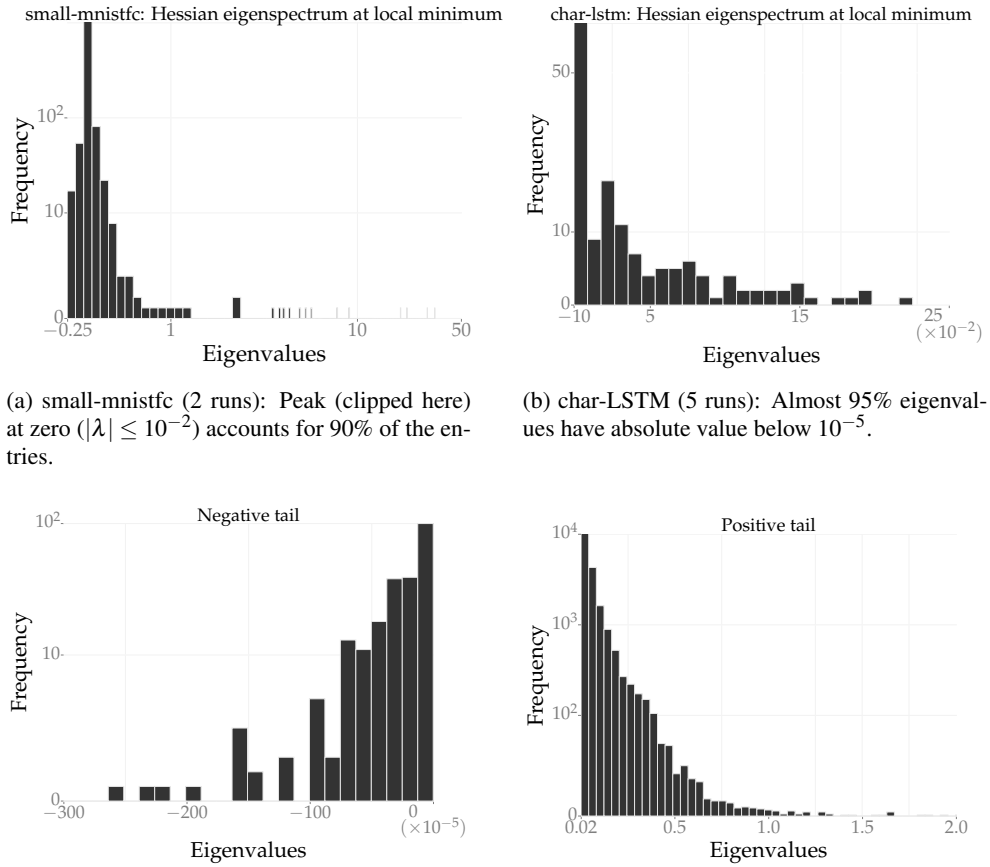

(a) small-mnistfc (2 runs): Peak (clipped here) at zero ($|\lambda| \leq 10^{-2}$) accounts for 90% of the entries.

(b) char-LSTM (5 runs): Almost 95% eigenvalues have absolute value below $10^{-5}$.

(c) Negative and positive eigenvalues of the Fisher information matrix of All-CNN-BN at a local minimum (4 independent runs). The origin has a large peak with $\approx 95\%$ near-zero ($|\lambda| \leq 10^{-5}$) eigenvalues (clipped here).

Figure 3: Universality of the Hessian: for a wide variety of network architectures, sizes and datasets, optima obtained by SGD are mostly flat (large peak near zero), they always have a few directions with large positive curvature (long positive tails). A very small fraction of directions have negative curvature, and the magnitude of this curvature is extremely small (short negative tails).

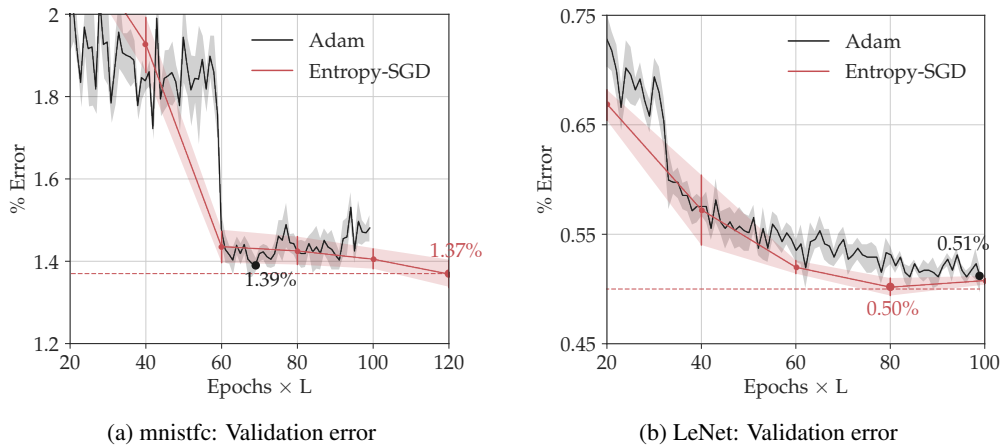

(a) mnistfc: Validation error

(b) LeNet: Validation error

Figure 4: Comparison of Entropy-SGD vs. Adam on MNIST

## 5.3 CIFAR-10

We train on CIFAR-10 without data augmentation after performing global contrast normalization and ZCA whitening (Goodfellow et al., 2013). We consider the All-CNN-C network of Springenberg et al. (2014) with the only difference that a batch normalization layer is added after each convolutional layer; all other architecture and hyper-parameters are kept the same. We train for 200 epochs with SGD and Nesterov's momentum during which the initial learning rate of 0.1 decreases by a factor of 5 after every 60 epochs. We obtain an average error of $7.71 \pm 0.19\%$ in 200 epochs vs. 9.08% error in 350 epochs that the authors in Springenberg et al. (2014) report and this is thus a very competitive baseline for this network. Let us note that the best result in the literature on non-augmented CIFAR-10 is the ELU-network by Clevert et al. (2015) with 6.55% test error.

We train Entropy-SGD with $L = 20$ for 10 epochs with the original dropout of 0.5. The initial learning rate of the outer loop is set to $\eta = 1$ and drops by a factor of 5 every 4 epochs, while the learning rate of the SGLD updates is fixed to $\eta' = 0.1$ with thermal noise $\varepsilon = 10^{-4}$. As the scoping scheme, we set the initial value of the scope to $\gamma_0 = 0.03$ which increases by a factor of 1.001 after each parameter update. Fig. 5 shows the training and validation error curves for Entropy-SGD compared with SGD. It shows that local entropy performs as well as SGD on a large CNN; we obtain a validation error of $7.81 \pm 0.09\%$ in about 160 effective epochs.

We see almost no plateauing of training loss or validation error for Entropy-SGD in Fig. 5a; this trait is shared across different networks and datasets in our experiments and is an indicator of the additional smoothness of the local entropy landscape coupled with a good scoping schedule for $\gamma$.

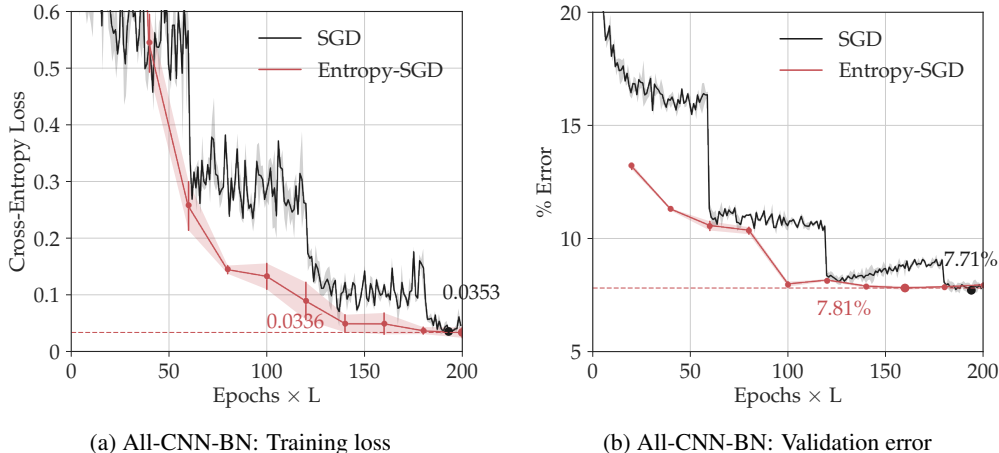

(a) All-CNN-BN: Training loss

(b) All-CNN-BN: Validation error

Figure 5: Comparison of Entropy-SGD vs. SGD on CIFAR-10

## 5.4 RECURRENT NEURAL NETWORKS

### 5.4.1 PENN TREE BANK

We train an LSTM network on the Penn Tree Bank (PTB) dataset for word-level text prediction. This dataset contains about one million words divided into a training set of about $930,000$ words, a validation set of $74,000$ words and $82,000$ words with a vocabulary of size $10,000$. Our network called PTB-LSTM consists of two layers with 1500 hidden units, each unrolled for 35 time steps; note that this is a large network with about 66 million weights. We recreated the training pipeline of Zaremba et al. (2014) for this network (SGD without momentum) and obtained a word perplexity of $81.43 \pm 0.2$ on the validation set and $78.6 \pm 0.26$ on the test set with this setup; these numbers closely match the results of the original authors.

We run Entropy-SGD on PTB-LSTM for 5 epochs with $L = 5$, note that this results in only 25 effective epochs. We do not use scoping for this network and instead fix $\gamma = 10^{-3}$. The initial learning rate of the outer loop is $\eta = 1$ which reduces by a factor of 10 at each epoch after the

third epoch. The SGLD learning rate is fixed to $\eta' = 1$ with $\varepsilon = 10^{-4}$. We obtain a word perplexity of $80.116 \pm 0.069$ on the validation set and $77.656 \pm 0.171$ on the test set. As Fig. 6a shows, Entropy-SGD trains significantly faster than SGD (25 effective epochs vs. 55 epochs of SGD) and also achieves a slightly better generalization perplexity.

### 5.4.2 CHAR-LSTM ON WAR AND PEACE

Next, we train an LSTM to perform character-level text-prediction. As a dataset, following the experiments of Karpathy et al. (2015), we use the text of War and Peace by Leo Tolstoy which contains about 3 million characters divided into training (80%), validation (10%) and test (10%) sets. We use an LSTM consisting of two layers of 128 hidden units unrolled for 50 time steps and a vocabulary of size 87. We train the baseline with Adam for 50 epochs with an initial learning rate of 0.002 that decays by a factor of 2 after every 5 epochs to obtain a validation perplexity of $1.224 \pm 0.008$ and a test perplexity of $1.226 \pm 0.01$.

As noted in Sec. 4.2, we can easily wrap Alg. 1 inside other variants of SGD such as Adam; this involves simply substituting the local entropy gradient in place of the usual back-propagated gradient. For this experiment, we constructed Entropy-Adam which is equivalent to Adam with the local entropy gradient (which is computed via SGLD). We run Entropy-Adam for 5 epochs with $L = 5$ and a fixed $\gamma = 0.01$ with an initial learning rate of 0.01 that decreases by a factor of 2 at each epoch. Note that this again results in only 25 effective epochs, i.e. half as much wall-clock time as SGD or Adam. We obtain a validation perplexity of $1.213 \pm 0.007$ and a test perplexity of $1.217 \pm 0.005$ over 4 independent runs which is better than the baseline. Fig. 6b shows the error curves for this experiment.

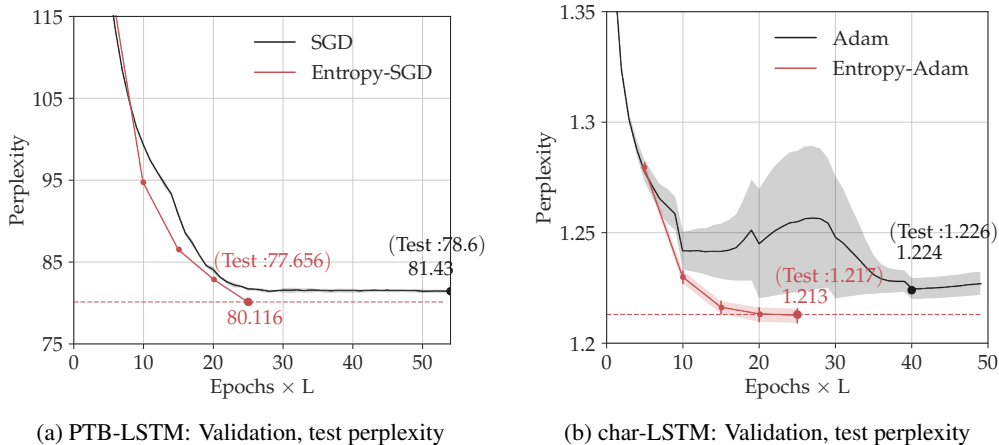

(a) PTB-LSTM: Validation, test perplexity  (b) char-LSTM: Validation, test perplexity

Figure 6: Comparison of Entropy-SGD vs. SGD / Adam on RNNs

| Model | Entropy-SGD | | SGD / Adam | |
|---|---|---|---|---|
| | Error (%) / Perplexity | Epochs | Error (%) / Perplexity | Epochs |
| mnistfc | $1.37 \pm 0.03$ | 120 | $1.39 \pm 0.03$ | 66 |
| LeNet | $0.5 \pm 0.01$ | 80 | $0.51 \pm 0.01$ | 100 |
| All-CNN-BN | $7.81 \pm 0.09$ | 160 | $7.71 \pm 0.19$ | 180 |
| PTB-LSTM | $77.656 \pm 0.171$ | 25 | $78.6 \pm 0.26$ | 55 |
| char-LSTM | $1.217 \pm 0.005$ | 25 | $1.226 \pm 0.01$ | 40 |

Table 1: Experimental results: Entropy-SGD vs. SGD / Adam

Tuning the momentum in Entropy-SGD was crucial to getting good results on RNNs. While the SGD baseline on PTB-LSTM does not use momentum (and in fact, does not train well with momentum) we used a momentum of 0.5 for Entropy-SGD. On the other hand, the baseline for char-LSTM was trained with Adam with $\beta_1 = 0.9$ ($\beta_1$ in Adam controls the moving average of the gradient) while we set $\beta_1 = 0.5$ for Entropy-Adam. In contrast to this observation about RNNs, all our experiments on CNNs used a momentum of 0.9. In order to investigate this difficulty, we monitored the angle between the local entropy gradient and the vanilla SGD gradient during training. This angle changes much more rapidly for RNNs than for CNNs which suggests a more rugged energy landscape for the former; local entropy gradient is highly uncorrelated with the SGD gradient in this case.

## 6   DISCUSSION

In our experiments, Entropy-SGD results in generalization error comparable to SGD, but always with lower cross-entropy loss on the training set. This suggests the following in the context of energy landscapes of deep networks. Roughly, wide valleys favored by Entropy-SGD are located deeper in the landscape with a lower empirical loss than local minima discovered by SGD where it presumably gets stuck. Such an interpretation is in contrast to theoretical models of deep networks (cf. Sec. 2) which predict multiple equivalent local minima with the same loss. Our work suggests that geometric properties of the energy landscape are crucial to generalize well and provides algorithmic approaches to exploit them. However, the literature lacks general results about the geometry of the loss functions of deep networks — convolutional neural networks in particular — and this is a promising direction for future work.

If we focus on the inner loop of the algorithm, SGLD updates compute the average gradient (with Langevin noise) in a neighborhood of the parameters while maintaining the Polyak average of the new parameters. Such an interpretation is very close to averaged SGD of Polyak & Juditsky (1992) and Bottou (2012) and worth further study. Our experiments show that while Entropy-SGD trains significantly faster than SGD for recurrent networks, it gets relatively minor gains in terms of wall-clock time for CNNs. Estimating the gradient of local entropy cheaply with few SGLD iterations, or by using a smaller network to estimate it in a teacher-student framework (Balan et al., 2015) is another avenue for extensions to this work.

## 7   CONCLUSIONS

We introduced an algorithm named Entropy-SGD for optimization of deep networks. This was motivated from the observation that the energy landscape near a local minimum discovered by SGD is almost flat for a wide variety of deep networks irrespective of their architecture, input data or training methods. We connected this observation to the concept of local entropy which we used to bias the optimization towards flat regions that have low generalization error. Our experiments showed that Entropy-SGD is applicable to large convolutional and recurrent deep networks used in practice.

## 8   ACKNOWLEDGMENTS

This work was supported by ONR N00014-13-1-034, AFOSR F9550-15-1-0229 and ARO W911NF-15-1-0564/66731-CS.

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

## A  STOCHASTIC GRADIENT LANGEVIN DYNAMICS (SGLD)

Local entropy in Def. (1) is an expectation over the entire configuration space $x \in \mathbb{R}^n$ and is hard to compute; we can however approximate its gradient using Markov chain Monte-Carlo (MCMC) techniques. In this section, we briefly review stochastic gradient Langevin dynamics (Welling & Teh, 2011) that is an MCMC algorithm designed to draw samples from a Bayesian posterior and scales to large datasets using mini-batch updates.

For a parameter vector $x \in \mathbb{R}^n$ with a prior distribution $p(x)$ and if the probability of generating a data item $\xi_k$ given a model parameterized by $x$ is $p(\xi_k | x)$, the posterior distribution of the parameters based on $N$ data items can be written as

$$p(x | \xi_{k \leq N}) \propto p(x) \prod_{k=1}^{N} p(\xi_k | x). \tag{10}$$

Langevin dynamics (Neal, 2011) injects Gaussian noise into maximum-a-posteriori (MAP) updates to prevent over-fitting the solution $x^*$ of the above equation. The updates can be written as

$$\Delta x_t = \frac{\eta}{2} \left( \nabla \log p(x_t) + \sum_{k=1}^{N} \nabla p(\xi_k | x_t) \right) + \sqrt{\eta}\ \varepsilon_t; \tag{11}$$

where $\varepsilon_t \sim \mathrm{N}(0, \varepsilon^2)$ is Gaussian noise and $\eta$ is the learning rate. In this form, Langevin dynamics faces two major hurdles for applications to large datasets. First, computing the gradient $\sum_{k=1}^{N} \nabla p(\xi_k | x_t)$ over all samples for each update $\Delta x_t$ becomes prohibitive. However, as Welling & Teh (2011) show, one can instead simply use the average gradient over $m$ data samples (mini-batch) as follows:

$$\Delta x_t = \frac{\eta_t}{2} \left( \nabla \log p(x_t) + \frac{N}{m} \sum_{k=1}^{m} \nabla p(\xi_k | x_t) \right) + \sqrt{\eta_t}\ \varepsilon_t. \tag{12}$$

Secondly, Langevin dynamics in (11) is the discrete-time approximation of a continuous-time stochastic differential equation (Mandt et al., 2016) thereby necessitating a Metropolis-Hastings (MH) rejection step (Roberts & Stramer, 2002) which again requires computing $p(\xi_k | x)$ over the entire dataset. However, if the learning rate $\eta_t \to 0$, we can also forgo the MH step (Chen et al., 2014). Welling & Teh (2011) also argue that the sequence of samples $x_t$ generated by updating (12) converges to the correct posterior (10) and one can hence compute the statistics of any function $g(x)$ of the parameters using these samples. Concretely, the posterior expectation $\mathbb{E}[g(x)]$ is given by $\mathbb{E}[g(x)] \approx \frac{\sum_{s=1}^{t} \eta_t\ g(x_t)}{\sum_{s=1}^{t} \eta_t}$; which is the average computed by weighing each sample by the corresponding learning rate in (12). In this paper, we will consider a uniform prior on the parameters $x$ and hence the first term in (12), viz., $\nabla \log p(x_t)$ vanishes.

Let us note that there is a variety of increasingly sophisticated MCMC algorithms applicable to our problem, e.g., Stochastic Gradient Hamiltonian Monte Carlo (SGHMC) by Chen et al. (2014) based on volume preserving flows in the "parameter-momentum" space, stochastic annealing thermostats (Santa) by Chen et al. (2015) etc. We can also employ these techniques, although we use SGLD for ease of implementation; the authors in Ma et al. (2015) provide an elaborate overview.

## B  PROOFS

*Proof of Lemma 2.* The gradient $-\nabla F(x)$ is computed in Sec. 4.1 to be $\gamma \left( x - \langle x';\ \Xi^\ell \rangle \right)$. Consider the term

$$x - \langle x';\ x \rangle = x - Z_{x,\gamma}^{-1} \int_{x'} x'\ e^{-f(x') - \frac{\gamma}{2}\ \|x - x'\|^2}\ dx'$$

$$\approx x - Z_{x,\gamma}^{-1} \int_s (x + s)\ e^{-f(x) - \nabla f(x)^\top s - \frac{1}{2}\ s^\top \left( \gamma + \nabla^2 f(x) \right) s}\ ds$$

$$= x \left( 1 - Z_{x,\gamma}^{-1} \int_s e^{-f(x) - \nabla f(x)^\top s - \frac{1}{2}\ s^\top \left( \gamma + \nabla^2 f(x) \right) s}\ ds \right) - Z_{x,\gamma}^{-1} \int_s s\ e^{-f(x) - \nabla f(x)^\top s - \frac{1}{2}\ s^\top \left( \gamma + \nabla^2 f(x) \right) s}\ ds$$

$$= -Z_{x,\gamma}^{-1}\ e^{-f(x)} \int_s s\ e^{-\nabla f(x)^\top s - \frac{1}{2}\ s^\top \left( \gamma + \nabla^2 f(x) \right) s}\ ds.$$

The above expression is the mean of a distribution $\propto e^{-\nabla f(x)^\top s - \frac{1}{2} s^\top (\gamma + \nabla^2 f(x)) s}$. We can approximate it using the saddle point method as the value of $s$ that minimizes the exponent to get

$$x - \langle x'; x \rangle \approx \left( \nabla^2 f(x) + \gamma I \right)^{-1} \nabla f(x).$$

Let us denote $A(x) := \left( I + \gamma^{-1} \nabla^2 f(x) \right)^{-1}$. Plugging this into the condition for smoothness, we have

$$\| \nabla F(x, \gamma) - \nabla F(y, \gamma) \| = \| A(x) \nabla f(x) - A(y) \nabla f(y) \|$$
$$\leq \left( \sup_x \| A(x) \| \right) \beta \, \| x - y \|.$$

Unfortunately, we can only get a uniform bound if we assume that for a small constant $c > 0$, no eigenvalue of $\nabla^2 f(x)$ lies in the set $[-2\gamma - c, c]$. This gives

$$\left( \sup_x \| A(x) \| \right) \leq \frac{1}{1 + \gamma^{-1} c}.$$

This shows that a smaller value of $\gamma$ results in a smoother energy landscape, except at places with very flat directions. The Lipschitz constant also decreases by the same factor. ∎

## C  CONNECTION TO VARIATIONAL INFERENCE

The fundamental motivations of (stochastic) variational inference (SVI) and local entropy are similar: they both aim to generalize well by constructing a distribution on the weight space. In this section, we explore whether they are related and how one might reconcile the theoretical and algorithmic implications of the local entropy objective with that of SVI.

Let $\Xi$ denote the entire dataset, $z$ denote the weights of a deep neural network and $x$ be the parameters of a variational distribution $q_x(z)$. The Evidence Lower Bound (ELBO) can be then be written as

$$\log p(\Xi) \geq \mathbb{E}_{z \sim q_x(z)} \left[ \log p(\Xi \mid z) \right] - \mathrm{KL} \left( q_x(z) \,\|\, p(z) \right); \tag{13}$$

where $p(z)$ denotes a parameter-free prior on the weights and controls, through their KL-divergence, how well the posited posterior $q_x(z)$ fits the data. Stochastic variational inference involves maximizing the right hand side of the above equation with respect to $x$ after choosing a suitable prior $p(z)$ and a family of distributions $q_x(z)$. These choices are typically dictated by the ease of sampling $z \sim q_x(z)$, e.g. a mean-field model where $q_x(z)$ factorizes over $z$, and being able to compute the KL-divergence term, e.g. a mixture of Gaussians.

On the other hand, if we define the loss as the log-likelihood of data, viz. $f(z) := -\log p(\Xi|z)$, we can write the logarithm of the local entropy in Eqn. (4) as

$$\log F(x, \gamma) = \log \int_{z \in \mathbb{R}^n} \exp \left[ -f(z; \Xi) - \frac{\gamma}{2} \| x - z \|^2 \right] \, dz,$$
$$\geq \int_{z \in \mathbb{R}^n} \left[ \log p(\Xi \mid z) - \frac{\gamma}{2} \| x - z \|^2 \right] \, dz; \tag{14}$$

by an application of Jensen's inequality. It is thus clear that Eqn. (13) and (14) are very different in general and one cannot choose a prior, or a variational family, that makes them equivalent and interpret local entropy as ELBO.

Eschewing rigor, formally, if we modify Eqn. (13) to allow the prior $p(z)$ to depend upon $x$, we can see that the two lower bounds above are equivalent iff $q_x(z)$ belongs to a "flat variational family", i.e. uniform distributions with $x$ as the mean and $p_x(z) \propto \exp \left( -\frac{\gamma}{2} \| x - z \|^2 \right)$. We emphasize that the distribution $p_x(z)$ depends on the parameters $x$ themselves and is thus, not really a prior, or one that can be derived using the ELBO.

This "moving prior" is absent in variational inference and indeed, a crucial feature of the local entropy objective. The gradient of local entropy in Eqn. (7) clarifies this point:

$$\nabla F(x, \gamma) = -\gamma \, (x - \langle z; \Xi \rangle) = -\gamma \, \mathbb{E}_{z \sim r(z;x)} \, [z];$$

where the distribution $r(z;x)$ is given by

$$r(z;\,x) \propto p(\Xi \mid z) \, \exp\left(-\frac{\gamma}{2} \, \|x-z\|^2\right);$$

it thus contains a data likelihood term along with a prior that "moves" along with the current iterate $x$.

Let us remark that methods in the deep learning literature that average the gradient through perturbations in the neighborhood of $x$ (Mobahi, 2016) or noisy activation functions (Gulcehre et al., 2016) can be interpreted as computing the data likelihood in ELBO (without the KL-term); such an averaging is thus different from local entropy.

## C.1 COMPARISON WITH SGLD

We use stochastic gradient Langevin dynamics (cf. Appendix A) to estimate the gradient of local entropy in Alg. 1. It is natural then, to ask the question whether vanilla SGLD performs as well as local entropy. To this end, we compare the performance of SGLD on two prototypical networks: LeNet on MNIST and All-CNN-BN on CIFAR-10. We follow the experiments in Welling & Teh (2011) and Chen et al. (2015) and set the learning rate schedule to be $\eta/(1+t)^b$ where the initial learning rate $\eta$ and $b$ are hyper-parameters. We make sure that other architectural aspects (dropout, batch-normalization) and regularization (weight decay) are consistent with the experiments in Sec. 5.

After a hyper-parameter search, we obtained a test error on LeNet of $0.63 \pm 0.1\%$ after 300 epochs and $9.89 \pm 0.11\%$ on All-CNN-BN after 500 epochs. Even if one were to disregard the slow convergence of SGLD, its generalization error is much worse than our experimental results; we get $0.50 \pm 0.01\%$ on LeNet and $7.81 \pm 0.09\%$ on All-CNN-BN with Entropy-SGD. For comparison, the authors in Chen et al. (2015) report 0.71% error on MNIST on a slightly larger network. Our results with local entropy on RNNs are much better than those reported in Gan et al. (2016) for SGLD. On the PTB dataset, we obtain a test perplexity of $77.656 \pm 0.171$ vs. 94.03 for the same model whereas we obtain a test perplexity of $1.213 \pm 0.007$ vs. 1.3375 for char-LSTM on the War and Peace dataset.

Training deep networks with SGLD, or other more sophisticated MCMC algorithms such as SGHMC, SGNHT etc. (Chen et al., 2014; Neal, 2011) to errors similar to those of SGD is difficult, and the lack of such results in the literature corroborates our experimental experience. Roughly speaking, local entropy is so effective because it operates on a transformation of the energy landscape that exploits entropic effects. Conventional MCMC techniques such as SGLD or Nose'-Hoover thermostats (Ding et al., 2014) can only trade energy for entropy via the temperature parameter which does not allow the direct use of the geometric information of the energy landscape and does not help with narrow minima.

