# Peer review of "Entropy-SGD: Biasing Gradient Descent Into Wide Valleys"

_ICLR 2017 — accepted_

[Public Comment · Martin Heusel · 08 Nov 2016]
**Relation to "Flat Minimum Search"**

Very interesting paper. The authors propose an algorithm which moves the parameters of a neural network towards flat landscapes of the error surface while decreasing the training error. I'm wondering what are the advantages over the related method "Flat Minima Search" (FMS) HS97. In HS97 the authors have shown a connection between flat minima and good generalization via MDL. They suggest the FMS algorithm to find such flat regions.
    

Reference:

[HS97] Hochreiter, S. and Schmidhuber, J. (1997), Flat Minima.

[Public Comment · Pratik A Chaudhari · 13 Nov 2016]
**updates to the paper**

We have updated the "Related work" section of the paper with a discussion of [HS97] and [KS16].

[HS97] Hochreiter, Sepp, and Jürgen Schmidhuber. "Flat minima." Neural Computation 9.1 (1997): 1-42.
[KS16] Keskar, Nitish Shirish, et al. "On Large-Batch Training for Deep Learning: Generalization Gap and Sharp Minima." arXiv:1609.04836 (2016).

[Official Review · AnonReviewer2 · rating 9 · confidence 3 · 16 Dec 2016 (modified: 16 Jan 2017)]
**Insightful Work**

This paper presents a principled approach to finding flat minima. The motivation to seek such minima is due to their better generalization ability. The idea is to add to the original loss function a new term that exploits both width and depth of the objective function. In fact, the regularization term can be interpreted as Gaussian convolution of the exponentiated loss. Therefore, the introduced regularization term is essentially Gaussian smoothed version of the exponentiated loss. The smoothing obviously tends to suppress sharp minima.

Overall, developing such regularization term based on thermodynamics concepts is very interesting. I have a couple of concerns that the authors may want to clarify in the rebuttal.

1. When reporting the generalization performance, the experiments report the number of epochs; showing the proposed algorithm reaches better generalization in fewer epochs than plain SGD. Is this the number of epochs it takes by line 7 of your algorithm, or it is the total number of epochs (line 3 and 7 all combined)? If the former, it is not a fair comparison. If you multiply the number of epochs of SGD (line 7) by the number iterations it takes to approximate Langevin dynamics, it seems you obtain little gain against plain SGD.

2. The proposed algorithm approximates the smoothed "exponentiated" loss (by smoothing I refer to convolution with the Gaussian). I am wondering how it compares against simpler idea of smoothing the original loss (dropping exponentiation)? Is the difference only in the motivation (e.g. thermodynamics interpretation) or it is deeper, e.g. the proposed scheme lends itself to more accurate approximation and/or achieves better generalization bound (in terms of the attained smoothness)? Smoothing the cost function without exponentiation allows simpler approximation (Monte Carlo integration instead of MCMC), e.g. see section 5.3 of

[Public Comment · ICLR 2017 conference · 16 Dec 2016]
**How is this better than stochastic-gradient Langevin dynamics, or stochastic variational inference?**

I agree with the motivations of the method, but this paper seems to be mostly re-inventing stochastic variational inference.  The SVI ELBO estimate also encourages the approximate posterior mean to head in directions that have high likelihood times volume.

More broadly, the marginal likelihood is also trying to achieve the same tradeoffs, which is the motivation for MCMC methods such as SGLD in the first place.  So It seems bizarre that you would compare Adam against a complicated method with SGLD in the inner loop, without comparing against the much simpler SGLD as well.

Finally, it's frustrating that you discuss free energy and entropy at length in words without giving their precise definitions.  If you did so, I suspect it would become clear that you're proposing maximizing the marginal likelihood, which is definitely a great idea but already well-known.  How does the entropy you optimize relate to the marginal likelihood and the variational ELBO?

[Official Review · AnonReviewer1 · rating 7 · confidence 4 · 18 Dec 2016 (modified: 20 Jan 2017)]
**official review**

The paper introduces a new regularization term which encourages the optimizer 
to search for a flat local minimum of reasonably low loss instead of seeking a 
sharp region of a low loss. This is motivated by some empirical observations that
local minima of good generalization performance tend to have flat shape. 
To achieve this, a regularization term based on the free local energy is proposed
and the gradient of this term, which do not have tractable closed-form solution, 
is obtained by performing Monte Carlo estimation using SGLD sampler. In the 
experiments, the authors show some evidence of the flatness of good local 
minima, and also the performance of the proposed method in comparison to the
Adam optimizer. 

The paper is well and clearly written. I enjoyed reading the paper. The connection
to the concept of free energy in optimization framework seems interesting. The 
motivation of pursuing flatness is also well analyzed with a few experiments. I'm
wondering if the first term in eqn. (8) is correct. I guess it should be f(x') not f(x)?
Also, I'm wondering why the authors did not add the experiment results on RNN in
the evaluation of the performance because char-lstm for text generation was 
already used for the flatness experiments. I think adding more experiments on 
various models and applications of deep architectures (e.g., RNN, seq2seq, etc.) 
will make the author's claim more persuasive. I also found the mixed usage of the
terminology, e.g., free energy and free entropy, a bit confusing.

[Public Comment · Csaba Szepesvari · rating 6 · confidence 4 · 01 Jan 2017]
**Smoothing the error surface to improve generalization; high level idea may be good, details are not entirely convincing.**

__Note__: An earlier version of the review (almost identical to the present one) for an earlier version of the paper (available on arXiV) can be found here:

[Official Review · AnonReviewer4 · rating 8 · confidence 4 · 02 Jan 2017 (modified: 19 Jan 2017)]
**Well-written paper explores promising direction for training generalizable deep neural network, but empirical results are too preliminary to support some arguments made**

Overview: 

This paper introduces a biasing term for SGD that, in theoretical results and a toy example, yields solutions with an approximately equal or lower generalization error. This comes at a computational cost of estimating the gradient of the biasing term for each iteration through stochastic gradient Langevin dynamics, approximating an MCMC sample of the log partition function of a modified Gibbs distribution. The cost is equivalent to adding an inner for-loop to the standard SGD algorithm for each minibatch.

Pros:
- Reviews and distills many results and theorems from past 2 decades that suggest a promising way forward for increasing the generalizability of deep neural networks
- Generally very well written and well presented results, with interesting discussion of eigenvalues of Hessian as a way to characterize “flat” minima
- Promising mathematical arguments suggest that E-SGD has generalization error bounded below by SGD, motivating further research in the area

Cons / points suggested for a rebuttal:
(1) One claim of the paper given in the abstract is ”experiments on competitive baselines demonstrate that Entropy-SGD leads to improved generalization and has the potential to accelerate training.“ This does not appear to be supported by the current set of experiments. As the authors comment in the discussion section, “In our experiments, Entropy-SGD results in a comparable generalization error as SGD, but always has a lower cross-entropy loss.” It's not clear to me how to reconcile those two claims.

(2) Similarly, the claim of accelerated training is not convincingly supported in the present version of the paper. Vanilla SGD requires a single forward pass through all M minibatches during one epoch for a parameter update, but the new method, E-SGD requires, L*M forward passes during one epoch where L is the number of Langevin updates, which require a minibatch sample each. This could in fact mean that E-SGD has worse computational complexity to reach the same point. In a remark on p.9, the authors note that a single epoch is defined to be “the number of parameter updates required to run through the dataset once.” It’s not clear to me how this answers the objection to a factor of L additional computations required for the inner-loop SGLD iterations. SGLD appears to introduces a potentially costly tradeoff that must be carefully managed by a user of E-SGD.

(3) As the previous two points suggest, the paper could use some attention to the magnitude of the claims. For example, the introduction reads “Actively biasing towards wide valleys aids generalization, in fact, we can optimize solely the free energy term to obtain similar generalization error as SGD on the original loss function.“ According the the values reported on pp.9-10, only on MNIST is the generalization error, using only the free energy term (the log partition function of the modified Gibbs distribution), equivalent to using only the SGD loss function. This corresponds to setting rho to 0 in equation (6). On CIFAR-10, rho = 0.01 is used.

(4) Another contribution of this paper, the characterization of the optimization landscape in terms of the eigenvalues of the Hessian and low generalization error being associated with flat local extrema, is helpful and interesting. I found the plots clear and useful. As another reviewer has already pointed out, there are high-level similarities to “Flat Minima” by Hochreiter and Schmidhuber (1997). The authors have responded already by adding a paragraph that helpfully explores some differences with H&S 1997. However, the similarities should also be carefully identified and mentioned. H&S 1997 includes detailed theoretical analysis that could be helpful for future work in this area, and has independently discovered a similar approach to training generalizable networks.

(5) It's not clear how the assumption about the eigenvalues that were made in section 4.4 / Appendix B affect the application of this result to real-world problems. What magnitude of c>0 needs to be chosen? Does this correspond to a measurable characteristic of the dataset? It's a little mysterious in the current version of the paper.

[Public Comment · Pratik A Chaudhari · 14 Jan 2017]
**Updates to the paper**

We thank the reviewers and the area chair for their insightful feedback. We have incorporated all comments into our current draft. We first discuss the updates to the paper and address common questions raised by the reviewers. We have also posted individual comments to the reviewers to address specific questions.

Updates
=====

a) We have updated the experimental section of the paper with new experiments on MNIST, CIFAR-10 and two datasets on RNNs (PTB and char-LSTM).

b) We have modified the algorithm to introduce a technique called "scoping". This increases the scope parameter \gamma as training progresses instead of fixing it and has the effect of exploring the parameter space in the beginning of training (Sec. 4.3). As a result of this, we can now train all our networks with only the local entropy loss instead of treating it as a regularizer (Eqn. 6).

c) For a fair comparison of the training time, we now plot the error curves in Figs. 4, 5 and 6 against the "effective number of epochs", i.e., the number of epochs of Entropy-SGD is multiplied by the number of Langevin iterations L (we set L=1 for SGD/Adam). Thus the x-axis is a direct measure of the wall-clock time agnostic to the underlying implementation and is proportional to the number of back-props as suggested.

We obtain significant speed-ups with respect to our earlier results due to scoping and the wall-clock training time for all our networks with Entropy-SGD is now comparable to SGD/Adam. In fact, Entropy-SGD is almost twice as fast as SGD on our experiments on RNNs and also obtains a better generalization error (cf. Fig. 6). The acceleration for CNNs on MNIST and CIFAR-10 is about 20%.

Table 1 (page 11) summarizes the experimental section of the paper.

d) Improved exposition of the algorithm in Sec. 4.2 that includes intuition for hyper-parameter tuning. We have expanded the discussion of experiments in Sec. 5.3, 5.4 to provide more details and insights that relate to the energy landscape of deep networks.

e) Appendix C discusses the possible connections to variational inference (this is the same material as the discussion with the AC below). Sec. C.1 presents an experimental comparison of local entropy vs. SGLD. We note here that our results using Entropy-SGD for both CNNs and RNNs are much better than vanilla SGLD in significantly smaller (~3-5x) wall-clock times.

Response to the reviewers:
================

>> smoothing of the original loss vs. local entropy
We discuss this in detail in the related work in Sec. 2 and Appendix C. While smoothing the original loss function using convolutions or averaging the gradient over perturbations of weights reduces the ruggedness of the energy landscape, it does not help with sharp, narrow valleys. Local entropy introduces a measure that focuses on wide local minima in the energy landscape (cf. Fig. 2 which has a "global" minimum at a wide valley); this is the primary reason for its efficacy. Smoothing the loss function can also, for instance, generate an artificial local minimum between two close by sharp valleys, which is detrimental to generalization.

>> unrealistic eigenvalue assumption in Sec. 4.3
We have clarified this point in Remark 4. Our analysis employs an assumption that the Hessian \nabla^2 f(x) does not have eigenvalues in the set [-2\gamma-c, c] for some c > 0. This is admittedly unrealistic, for instance, the eigenspectrum of the Hessian in Fig. 1 has a large fraction of its eigenvalues almost zero. Let us note though that Fig. 1 is plotted at a local minimum, from our experiments, the eigenspectrum is less sparse in the beginning of training.

We would like to remark that the bound on uniform stability in Thm. 3 by Hardt et al. assumes global conditions on the smoothness of the loss function; one imagines that Eqn. 9 remains qualitatively the same (in particular, with respect to the number of training iterations) even if this assumption is violated to an extent before convergence happens. Obtaining a rigorous generalization bound without this assumption requires a dynamical analysis of SGD and seems out of reach currently.

[Final Decision · Program Chairs · 06 Feb 2017]
**ICLR committee final decision**

This paper presents both an analysis of neural net optimization landscapes, and an optimization algorithm that encourages movement in directions of high entropy. The motivation is based on intuitions from physics.
 
 Pros
  - the main idea is well-motivated, from a non-standard perspective.
  - There are lots of side-experiments supporting the claims for the motivation.
 Cons
  - The propose method is very complicated, and it sounds like good performance depended on adding and annealing yet another hyperparameter, referred to as 'scoping'.
  - The motivating intuition has been around for a long time in different forms. In particular, the proposed method is very closely related to stochastic variational inference, or MCMC methods. Appendix C makes it clear that the two methods aren't identical, but I wish the authors had simply run SVI with their proposed modification, instead of appearing to re-invent the idea of maximizing local volume from scratch. The intuition that good generalization comes from regions of high volume is also exactly what Bayes rule says.
 
 In summary, while there is improvement for the paper, the idea is well-motivated and the experimental results are sound.